# Molecular Mechanisms of Thyroid Hormone Signaling in Thyroid Cancer: Oncogenesis, Progression, and Therapeutic Implications

**DOI:** 10.3390/biomedicines13102552

**Published:** 2025-10-20

**Authors:** Changhao Zhou, Wei Liu, Jiaojiao Zheng, Qiao Wu, Zhilong Ai

**Affiliations:** 1Department of Surgery (Thyroid & Breast), Zhongshan Hospital, Fudan University, Shanghai 200030, China; 22301050172@m.fudan.edu.cn (C.Z.); liu.wei6@zs-hospital.sh.cn (W.L.); zheng.jiaojiao@zs-hospital.sh.cn (J.Z.); 2School of Basic Medical Sciences, Fudan University, Shanghai 200030, China

**Keywords:** thyroid neoplasms, thyroid hormones, integrin αvβ3, TSH suppression therapy, thyroid hormone receptor β

## Abstract

Thyroid cancer, as a highly hormone-dependent malignancy, is significantly regulated by thyroid hormones (T3/T4) and thyroid-stimulating hormone (TSH) signaling in its initiation and progression. This article comprehensively reviews the roles of thyroid hormones and their regulatory factor TSH in thyroid carcinogenesis and development, addressing related research from molecular mechanisms and clinical correlations to therapeutic strategies. It focuses on elucidating the impact of key mechanisms—such as elevated integrin αvβ3 expression and TRβ receptor mutations under hyperthyroid or hypothyroid conditions—on tumor progression. Furthermore, it evaluates the clinical utility and potential risks of TSH suppression therapy in patients stratified by risk, aiming to provide a theoretical basis for optimizing individualized treatment strategies.

## 1. Introduction

The thyroid hormone (TH) plays a critical role in regulating metabolism, growth, and development throughout the organism. It enhances mitochondrial oxidative phosphorylation, thereby boosting ATP production and increasing oxygen consumption rates, which collectively contribute to an elevation in the basal metabolic rate [1]. In lipid metabolism, thyroid hormone stimulates the activity of lipases, thereby promoting lipolysis. Concurrently, it accelerates cholesterol synthesis and enhances its conversion and subsequent excretion as bile acids [2]. In glucose metabolism, TH not only improves intestinal glucose absorption but also promotes hepatic glycogenolysis and gluconeogenesis, thereby augmenting systemic glucose utilization. Disruption of thyroid function profoundly disturbs the homeostasis of these metabolic processes. Hyperthyroidism—a condition characterized by excessive synthesis and secretion of thyroid hormones—results in a generalized state of hypermetabolism. Key clinical manifestations include heat intolerance, profuse sweating, unintentional weight loss, tachycardia, and increased nervous excitability [3,4]. Hypothyroidism is characterized by reduced synthesis and secretion of thyroid hormones, leading to a generalized decline in metabolic activity. Key manifestations include a decreased metabolic rate, diminished caloric expenditure, mental and physical fatigue, and impaired lipolysis and gluconeogenesis [5].

The hypothalamic–pituitary–thyroid axis (HPT axis) constitutes a key neuroendocrine system responsible for the regulation of TH production(Figure 1). The process is initiated in the hypothalamus with the secretion of the thyrotropin-releasing hormone (TRH). The TRH is transported via the hypothalamic–pituitary portal circulation to the anterior pituitary, where it stimulates the release of the thyroid-stimulating hormone (TSH). The TSH enters the systemic circulation and acts directly on the thyroid gland, binding to receptors on thyroid follicular cells to promote the synthesis and secretion of TH. Finally, thyroid hormones are released into the bloodstream to exert their effects on peripheral target tissues [6]. Elevated circulating TH levels trigger a negative feedback loop that suppresses the secretion of TRH from the hypothalamus and inhibits TSH release from the anterior pituitary. This inhibition reduces the stimulation of the thyroid gland, thereby decreasing the synthesis and secretion of TH and preventing hormonal excess. This closed-loop regulatory mechanism is essential for maintaining the homeostatic balance of thyroid hormone levels in the bloodstream [7].

The hypothalamus releases TRH, which stimulates thyrotrope cells in the pituitary gland to synthesize and release the TSH. The TSH then acts on the thyroid gland to promote the production and secretion of thyroid hormones T3 and T4. T3 and T4 exert negative feedback inhibition on both the pituitary and the hypothalamus to suppress the release of TSH and TRH, respectively, completing the feedback loop and maintaining hormonal homeostasis. Green arrows represent stimulatory pathways; red arrows represent inhibitory feedback.

Thyroid cancer (TC) is the most common endocrine malignancy worldwide, accounting for approximately 2.1% of all cancer diagnoses globally [8]. Over the past three decades, the global incidence of thyroid cancer has experienced a substantial increase, with annual growth rates ranging from 3% to 7% [9,10]. According to epidemiological data, the age-standardized incidence rate (ASR) of thyroid cancer worldwide is 6.1/100,000, with significant gender differences. The ASR for women (9.1/100,000) is approximately three times that of men (3.1/100,000) [11,12].

According to the World Health Organization (WHO) 2022 publication WHO Classification of Tumours of Endocrine Organs (5th edition), thyroid pathologies encompass a spectrum of disorders ranging from benign follicular nodular diseases to malignant thyroid neoplasms [13]. Regarding histological classification, thyroid cancer can be divided into four categories: papillary thyroid carcinoma (PTC), follicular thyroid carcinoma (FTC), Medullary thyroid carcinoma (MTC) and anaplastic thyroid carcinoma (ATC) [14]. Differentiated thyroid carcinoma (DTC), representing approximately 90% of all thyroid malignancies, primarily comprises PTC and FTC [15]. DTCs are frequently driven by mutations in the MAPK pathway, typically at BRAF or RAS oncoproteins [16]. MTC originated from parafollicular C-cells and is strongly related to germline or somatic mutations in the *RET* proto-oncogene and is less frequently associated with *KRAS* and *BRAF*. And its biological behavior, genetic basis, and lack of responsiveness to TSH are fundamentally distinct. [17]. ATC is derived from follicular thyroid cells and carries the highest risk of death, yet accounts for only a small fraction of all TC cases [18]. This review will focus primarily on DTC, given its high prevalence and the well-established role of thyroid hormone signaling in its pathogenesis, though relevant contrasts with MTC and ATC will be noted where appropriate.

Beyond genetic alterations, emerging evidence underscores the importance of epigenetic modifications in thyroid tumorigenesis. These mechanisms primarily involve hypermethylation of DNA in promoter CpG islands and post-translational modifications of histones (such as deacetylation and methylation), leading to transcriptional silencing of key genes [19,20,21]. In thyroid cancer, this results in the loss of thyroid-specific functions, such as impaired radioactive iodine uptake due to silencing of the NIS and TSHR genes [22]. These mechanisms can promote uncontrolled cell proliferation through inactivation of tumor suppressor genes like *RASSF1A*, *PTEN*, and *RAP1GAP* [23,24,25]. The reversibility of these epigenetic marks offers promising therapeutic avenues. Strategies aim to reverse gene silencing through demethylating agents and histone deacetylase inhibitors (HDACi) to restore differentiation and suppress tumor growth, particularly in advanced, refractory cases [26].

Thyroid hormones precisely regulate metabolic homeostasis via the HPT axis, and their dysregulation is a direct cause of systemic pathologies such as hyperthyroidism and hypothyroidism. Thyroid cancer, which arises from the follicular epithelial cells of the thyroid gland, has been closely linked to T3 and T4 signaling pathways in both its pathogenesis and progression. At physiological levels, T3 can promote the expression of differentiation-related genes through TRβ activation and exert an anti-proliferative influence on tumor growth [27]. Conversely, clinical evidence indicates that patients with hyperthyroidism exhibit an elevated risk of developing thyroid cancer compared to individuals with normal thyroid function [28]. These observations highlight the paradoxical duality of thyroid hormones in thyroid pathophysiology. This review aims to provide a systematic integrative analysis of the duality of T3/T4 signaling in thyroid cancer. We will integrate evidence from two opposing pathways within this system: non-genomic signaling mediated by integrin αvβ3 and genomic signaling mediated by TRβ. By elucidating the interactions between these pathways, this study aims to establish an innovative conceptual framework that not only clarifies disease mechanisms but also reveals untapped therapeutic potential.

## 2. The Molecular Biology of T3/T4 and Thyroid Cancer

Thyroid hormones are key signaling molecules that regulate metabolism, development, and cell proliferation in the body. Their classic action depends on the genomic effects mediated by nuclear receptors (TRα/TRβ): the hormone-receptor complex binds to the thyroid hormone response element (TRE) in the promoter region of target genes, regulating the transcription of genes related to the cell cycle and differentiation. Notably, the TRβ subtype is highly expressed in thyroid tissue, and its activation can inhibit tumor proliferation and promote differentiation, thereby exerting an anticancer effect [29].

Within the tumor microenvironment, thyroid hormones contribute to cancer progression via nongenomic signaling pathways, primarily through their interaction with the cell membrane-bound integrin receptor αvβ3 [30]. αvβ3, as a transmembrane dimeric protein, exhibits significantly elevated expression levels in malignant tumors. The binding of T4/T3 to αvβ3 bypasses classical nuclear receptors and directly activates downstream oncogenic signaling cascades. This pathway mediates signal transduction through the dual axes of MAPK/ERK and PI3K/Akt [31]. It further regulates malignant phenotypes such as cell proliferation, angiogenesis, metastasis, apoptosis resistance, and immune evasion, becoming a key driver of thyroid cancer progression.

### 2.1. Integrin αvβ3-Mediated Carcinogenic Signaling in Thyroid Cancer

Integrin αvβ3 is a transmembrane dimeric protein on the cell membrane that binds thyroid hormones [32] and plays an important regulatory role in the interaction between cells and extracellular matrix proteins [33]. Thyroid hormones bind to integrin αvβ3 on the cell membrane, a process independent of classical nuclear TRs. This interaction triggers two intracellular signaling cascades: the MAPK/ERK and PI3K/Akt pathways, which modulate diverse physiological processes. Structurally, integrin αvβ3 contains two distinct hormone-binding domains: S1 and S2. The S1 domain exhibits high specificity for T3 and transduces signals via Src kinase to activate the PI3K/Akt pathway. In contrast, the S2 domain primarily binds T4, though it also interacts with T3, leading to activation of the MAPK/ERK signaling cascade [30]. The co-activation of these pathways serves as a central signaling hub, coordinating multiple tumorigenic processes.

#### 2.1.1. Activation of MAPK.ERK and PI3K/Akt Pathways

MAPK/ERK pathway activation: At physiological concentrations, T4 exerts a significant proliferative effect via the αvβ3 pathway, whereas the action of T3 under similar conditions is relatively more complex and less pronounced. Upon binding to the S2 site of the αvβ3 receptor, T4 first initiates the activation of Src kinase. The activated c-Src subsequently phosphorylates and activates Ras, which in turn stimulates Raf. Phosphorylated Raf then activates MEK1/2, ultimately leading to the phosphorylation and activation of ERK1/2 [30,32]. Phosphorylated ERK1/2 is subsequently transported to the cell nucleus, and it activates transcription factors such as Elk-1, c-Fos, and c-Myc [34].

PI3K/Akt pathway activation: Upon binding to the S1 site of integrin αvβ3, T3 activates PI3K, which catalyzes the conversion of PIP2 to PIP3 [30]. PIP3 then recruits Akt to the plasma membrane, where it is phosphorylated and activated [35,36]. The activated Akt subsequently enhances cell survival, proliferation, and metabolic processes by phosphorylating key downstream targets, including mTOR, FOXO, PTEN and GSK-3β [37,38].

#### 2.1.2. Promotion of Proliferation and Cell Cycle Progression

The proliferative signal is predominantly driven by the MAPK/ERK axis. Phosphorylated ERK1/2 and Akt activate downstream transcription factors, including Elk-1, c-Fos, and c-Myc. These factors translocate into the nucleus and drive the expression of early-response genes such as *FOS*, *JUN*, and *MYC* and significantly upregulate Cyclin D1 expression [32]. This T4-αvβ3-ERK-Cyclin D1 axis facilitates cells to enter the S phase from the G1 phase, thereby promoting uncontrolled proliferation [39].

#### 2.1.3. Induction of Angiogenesis

The thyroid gland is a central component of the endocrine system, secreting thyroid hormones that are distributed systemically via the bloodstream. Consequently, the thyroid is highly vascularized, being supplied by two major arteries and drained by three major veins. The promotion of angiogenesis by thyroid hormones through the αvβ3 signaling pathway is mechanistically linked to tumor development and progression [40]. Tumor cells commonly exist in a hypoxic state and generate excessive levels of reactive oxygen species (ROS), resulting in aberrant metabolic activity and intracellular acidosis [41]. Therefore, angiogenesis plays a critical role in facilitating nutrient delivery to support tumor survival and expansion. Furthermore, it promotes hematogenous metastasis—a key route of tumor dissemination—by enhancing vascular permeability and providing avenues for invasive tumor cells to enter the circulation [42].

The pro-angiogenic effect of T4 was experimentally confirmed using the chick chorioallantoic membrane (CAM) assay [43]. The underlying mechanism involves the activation of both MAPK/ERK and PI3K/Akt signaling pathways by T3 and T4. These pathways act synergistically to enhance the transcriptional activity of HIF-1α [44]. The specific mechanism involves Akt-mediated inhibition of GSK-3β and activation of mTOR, which both impede the prolyl hydroxylase (PHD)-dependent hydroxylation of HIF-1α. Under physiological conditions, hydroxylated HIF-1α is targeted for ubiquitination and subsequent proteasomal degradation. However, Akt/mTOR signaling disrupts this process, resulting in aberrant stabilization and accumulation of HIF-1α protein even in normoxia. As the master transcriptional regulator of angiogenesis, HIF-1α drives the expression of vascular endothelial growth factor (VEGF) and its principal receptor VEGFR2 [45]. It is hypothesized that VEGF overexpression is a hallmark feature of DTC, contributing to enhanced tumor growth and invasiveness [46,47]. Furthermore, thyroid hormones modulate the transcription of multiple tumor-associated genes (Table 1) —such as *HIF1A*, *VEGF*, *FGF2*, *PDGF*, *EGFR*, *MMP*, and *NOS2*—all of which are implicated in the regulation of angiogenesis [48,49].

#### 2.1.4. Enhancement of Tumor Metastasis and Invasion

The MAPK/ERK signaling pathway regulates the transcription of matrix metalloproteinase (MMP) genes, notably MMP2 and MMP9, thereby modulating their expression levels [60]. Matrix metalloproteinases are zinc-dependent endopeptidases that exhibit proteolytic activity through zinc ions at their catalytic sites. Specifically, MMP-2 and MMP-9 facilitate tumor metastasis by degrading components of the extracellular matrix, enabling the dissociation of cancer cells from the primary tumor and promoting invasive behavior [60,61]. Additionally, MMP-2 and MMP-9 facilitate tumor invasion and angiogenesis through the proteolytic activation of TGF-β [62]. Current evidence indicates that MMPs contribute to multiple stages of the metastatic cascade, including primary tumor growth, angiogenesis, intravasation, and extravasation [59].

Rho GTPases constitute a family of small GTP-binding proteins within the Ras superfamily and function as critical molecular switches that regulate a wide range of cellular processes in eukaryotic cells [63]. Members of the Rho GTPase family—primarily Rac1, Cdc42, and RhoA—play pivotal roles in regulating tumor cell motility. Activation of the αvβ3 signaling pathway stimulates PI3K/Akt, leading to Akt phosphorylation and subsequent activation of Rac1/Cdc42 guanine nucleotide exchange factors (GEFs), such as Tiam1 and Vav2. This cascade promotes the formation of lamellipodia and filopodia at the leading edge of tumor cells, thereby enhancing their motility and directional migration [64]. The MAPK/ERK pathway regulates RhoA activity, which in turn drives the assembly of contractile stress fibers at the cell rear, thereby generating the mechanical force required for cellular migration [63,65,66,67].

#### 2.1.5. Inhibition of Apoptosis

Resveratrol (RV) is a natural compound known to induce apoptosis in various cancer cell types. Treatment of thyroid cancer cells with RV leads to significant upregulation of *p21*, *FOS*, and *JUN* mRNA expression, indicating activation of pro-apoptotic signaling pathways [68]. After T4 treatment, the expression of these pro-apoptotic markers was suppressed, demonstrating an anti-apoptotic effect of thyroxine. This action is mediated through T4 binding to integrin αvβ3, which subsequently activates the MAPK/ERK signaling cascade. Activated ERK1/2 directly or indirectly phosphorylates critical residues (Ser112 and Ser136) on the pro-apoptotic protein BAD, thereby inhibiting its cell death-promoting activity [69]. This phosphorylation inactivates BAD and sequesters it away from its binding partners. As a result, Bcl-2 remains active and exerts its anti-apoptotic function by stabilizing the mitochondrial outer membrane, thereby preventing the release of cytochrome c. Consequently, the initiation of caspase-9 activation is inhibited, effectively blocking the mitochondrial apoptotic pathway [70,71].

#### 2.1.6. Facilitation of Immune Evasion

Programmed death-ligand 1 (PD-L1), also referred to as CD274, is a cell surface glycoprotein member of the B7 family. Elevated PD-L1 expression on tumor cells engages with programmed death-1 (PD-1) receptors on T cells, delivering inhibitory signals that result in T cell exhaustion, suppression of proliferation, diminished cytokine secretion, and apoptosis. Consequently, this mechanism facilitates immune evasion by tumor cells [72].

T4 upregulates the expression of the critical downstream effector HIF-1α via the integrin αvβ3 pathway. As a transcription factor, HIF-1α translocates into the nucleus, dimerizes with HIF-1β, and binds to hypoxia-response elements (HREs) within the promoter regions of target genes [73]. A key target of this regulatory axis is PD-L1. HIF-1α potently induces PD-L1 expression on tumor cells, resulting in robust suppression of T cell activity and subsequent inhibition of the anti-tumor immune response [74].

#### 2.1.7. The Effects of Tetraiodothyronine (Tetrac)

Tetrac, a deaminated derivative of T4, functions as a structural analog that specifically targets the S1 binding site on integrin αvβ3. By competitively inhibiting T4 binding to αvβ3, Tetrac attenuates downstream ERK phosphorylation and suppresses thyroid cancer cell proliferation [75]. In MCT cells xenografted onto the chick chorioallantoic membrane (CAM) model, treatment with tetrac resulted in a reduction of over 60% in both tumor volume and hemoglobin content [76]. Tetrac and its nanoformulation, nano-diamino-tetrac (NDAT), exhibit promising potential as anti-metastatic agents. NDAT enhances sustained drug action, specifically targets the tumor microenvironment, and inhibits MMP activity, demonstrating significant therapeutic value in suppressing the progression of advanced tumors [77].

#### 2.1.8. Novel Targeted Therapeutic Strategies Against MAPK/ERK and PI3K/Akt Pathways

For radioactive iodine-refractory and advanced thyroid cancer, targeted therapies against the MAPK and PI3K/Akt pathways are a treatment cornerstone [78]. Clinical evidence shows that inhibiting key nodes in these pathways effectively suppresses tumor progression and improves outcomes.

Within the MAPK pathway, the BRAF V6000E mutation is a pivotal therapeutic target. Selective BRAF V6000E inhibitors vemurafenib and dabrafenib have demonstrated significant antitumor activity in clinical trials [79]. For ATC, the combination of dabrafenib and the MEK inhibitor trametinib has shown breakthrough efficacy, achieving a 69% ORR in a phase II study and subsequently gaining FDA approval as the first effective targeted regimen for this disease [80]. Emerging research frontiers are increasingly centered on exploring synergistic combinations of targeted therapy and immunotherapy. Preclinical studies demonstrate that co-administration of BRAF V600E inhibitors with anti-PD-1/PD-L1 antibodies significantly suppresses ATC progression and remodels the tumor immune microenvironment toward an immunologically active state [81,82]. Although clinical validation is still evolving, this combined approach represents a pivotal strategy for overcoming therapeutic resistance and improving outcomes in refractory thyroid cancer.

### 2.2. The Tumor Suppressive Effect of TRβ

TRs belong to the nuclear receptor superfamily [83] and are encoded by two distinct genes: THRA on chromosome 17 and THRB on chromosome 3, which give rise to the α and β subtypes, respectively. Functional TR isoforms include TRα1, TRβ1, TRβ2, and TRβ3, all of which contain ligand-binding domains (LBDs) that specifically bind T3. In contrast, the TRα2 isoform lacks a functional LBD and is incapable of ligand binding [84].

The v-erbA oncogene, derived from the avian erythroblastosis virus (AEV), shares 82% amino acid sequence identity with chicken TRα1 and demonstrates high structural and functional homology [85]. In contrast to TRα1, v-erbA lacks a functional LBD and is unable to bind T3 [86]; however, it retains an intact DNA-binding domain, enabling it to bind thyroid hormone response elements (TREs). v-erbA competes with wild-type TRα1 and TRβ for TRE occupancy, thereby inhibiting their DNA-binding activity [87]. Furthermore, v-erbA recruits nuclear receptor co-repressors (NCoR and SMRT), leading to constitutive repression of target gene transcription [88]. This aberrant repression disrupts the normal regulation of cellular differentiation by TRs and promotes malignant proliferation [89].

LBD of TRβ plays a crucial role in tumor suppression and requires ligand activation to transition into a transcriptionally active conformation capable of inhibiting cancer progression. Structurally, the LBD consists of 12 α-helices (H1–H12) that form a hydrophobic ligand-binding pocket. Helix H12 is particularly essential for co-regulator recruitment. In the unliganded state, H12 adopts a position parallel to the ligand-binding pocket, exposing a binding interface for co-repressors on helices H3–H5. Upon T3 binding, H12 undergoes a 60° rotation, sealing the ligand pocket and forming a binding surface for co-activators [88]. Thus, the LBD functions as a molecular switch whose T3-binding status determines the identity of associated co-regulators, ultimately directing opposing transcriptional outcomes. Upon activation by T3, TRβ recruits co-activators such as the histone acetyltransferases p300/CBP through its LBD. This complex facilitates chromatin relaxation and activates the expression of tumor suppressor genes, including *p21*, *DIO2*, and *BTG2* [29].

#### 2.2.1. The Function of Mutant TRβ

In clear cell renal cell carcinoma (ccRCC), a deletion in chromosome 3p was observed in 75% of cases, with TRβ gene mutations present in 32% of these tumors [90]. In hepatocellular carcinoma (HCC), truncation mutations affecting TRα1 and TRβ were identified in 53% of cases, while point mutations in LBD were detected in 65–76% of patients [91]. In thyroid cancer, studies indicate significantly reduced TRβ1 mRNA expression in both papillary and follicular thyroid carcinomas, along with aberrant TRα1 and TRβ1 protein expression in tumor tissues [92]. Dr. Sheue-yann Cheng and her team employed transgenic technology to generate a *Thrb^PV/PV^* knock-in mouse model, which spontaneously develops thyroid tumors that closely mimic the pathological progression observed in human thyroid cancer [93]. These findings demonstrate that deficiency or mutation of the thyroid hormone receptor β significantly elevates cancer susceptibility.

The oncogenic mechanism of mutant TRβ is largely attributed to its dominant-negative activity [94]. Under normal conditions, unliganded wild-type TRβ recruits co-repressor complexes, including NCo/SMRT and HDACs, to suppress target gene transcription [95]. Upon binding to T3, TRβ undergoes a conformational change, releasing co-repressors and instead recruiting co-activators, such as SRC and p300/CBP, thereby initiating transcription [96]. Most TRβ mutations are localized to the LBD and hinge region, resulting in a structurally impaired LBD that cannot be properly activated by T3 [97]. Consequently, mutant TRβ retains strong co-repressor binding capacity, leading to sustained repression of tumor-suppressive genes and ultimately promoting tumor development.

#### 2.2.2. PPARγ/PPRE Pathway

Peroxisome Proliferator-Activated Receptor Gamma (PPARγ) is a nuclear receptor that regulates adipocyte differentiation, glucose and lipid metabolism, and cellular differentiation [98]. In thyroid cancer, PPARγ activation promotes the transcription of tumor suppressor genes such as *p21* and *PTEN*, thereby inhibiting proliferation and differentiation, and exerting anti-tumor effects [99]. Wild-type TRβ acts synergistically with PPARγ: both are co-expressed in thyroid follicular cells, and TRβ enhances PPARγ-mediated transcription by directly binding to PPAR response elements (PPREs). In contrast, mutant TRβ exerts a dominant-negative effect over PPARγ [91]; it retains DNA-binding ability, competes for PPRE occupancy, and disrupts transcriptional activation of tumor-suppressive programs [100].

#### 2.2.3. PI3K/Akt Pathway

TRβ critically regulates the oncogenic PI3K/Akt pathway in a ligand-dependent manner. Wild-type TRβ and mutant TRβ exert opposing regulatory effects on this pathway: wild-type TRβ promotes the dephosphorylation of Akt via protein phosphatase 2A (PP2A), thereby inhibiting Akt activity, suppressing cancer cell proliferation and migration, and exerting anti-tumor effects [101]. In contrast, LBD of mutant TRβ interacts with the C-terminal SH2 domain of the PI3K regulatory subunit p85α, leading to enhanced cell proliferation, survival, motility, and inhibition of apoptosis—thereby promoting oncogenesis [101].

#### 2.2.4. Integrin-Src-FAK Pathway

The integrin–Src–FAK signaling pathway plays a key role in cellular signal transduction and the regulation of physiological processes. Focal adhesion kinase (FAK), a non-receptor tyrosine kinase, autophosphorylates at Tyr397 and recruits Src via its SH2 domain. This pathway enhances tumor invasiveness by facilitating epithelial–mesenchymal transition (EMT) and upregulating MMPs such as MMP-2 and MMP-9 [102,103]. Mutant TRβ can directly bind to integrin α5β1, leading to activation of the FAK–Src cascade and promotion of tumor metastasis. Suzuki et al. reported a fivefold increase in MMP-9 expression in thyroid tumors of *Thrb^PV/PV^* mice [102], providing direct evidence that mutant TRβ drives metastatic behavior.

#### 2.2.5. TRβ-*RUNX2*

*RUNX2* is one of three members of the RUNX transcription factor family in the human genome (*RUNX1*, *RUNX2*, *RUNX3*) [104], and is characterized by a conserved Runt homology domain [105]. Aberrant expression of *RUNX2* drives oncogenesis by stimulating angiogenesis [106], promoting metastasis [107], and enhancing drug resistance [108]. Wild-type TRβ binds to negative thyroid hormone response elements (nTREs) within the *RUNX2*-P1 promoter region [109], and recruits nuclear receptor co-repressors such as NCoR and SMRT [110], thereby repressing *RUNX2* transcription. In contrast, mutant TRβ fails to respond to T3 due to structural alterations and is unable to effectively suppress *RUNX2* expression, thereby contributing to its oncogenic activity.

#### 2.2.6. Prognostic Significance of TRβ Mutations

The presence of TRβ mutations or loss of expression is a significant marker of poor prognosis in thyroid cancer. Functionally impaired TRβ mutants, such as the dominant-negative TRβPV, not only lose transcriptional activity but also drive tumor progression through extranuclear signaling mechanisms [93]. For instance, TRβPV physically interacts with the p85α regulatory subunit of PI3K, leading to constitutive activation of the PI3K/Akt pathway, which enhances cell proliferation and suppresses apoptosis [101,111].

Moreover, TRβPV disrupts proteasomal degradation pathways, resulting in aberrant accumulation of oncoproteins such as PTTG and β-catenin [112]. This leads to mitotic abnormalities, chromosomal instability, and epithelial–mesenchymal transition (EMT)-like phenotypes, further driving dedifferentiation and metastatic behavior [113].

In summary, TRβ mutations enhance tumor invasiveness, including extraluminal invasion, vascular invasion, and distant metastasis [114]. Furthermore, these mutations may confer resistance to conventional therapies such as TSH suppression and radioactive iodine treatment, highlighting their potential as prognostic biomarkers and therapeutic targets in advanced thyroid cancer [115].

#### 2.2.7. Preclinical Mechanism Studies of TRβ in Antitumor Therapy

Preclinical studies have established the potent tumor-suppressive mechanisms of TRβ agonists in ATC [116]. TRβ agonists GC-1 reprogram the transcriptome and epigenome, leading to reduced cell proliferation, and metastasis, while promoting redifferentiation and apoptosis [117]. Key mechanisms include the induction of a tumor-suppressive gene network involving thyroid differentiation markers (e.g., NIS/SLC5A5, TG, PAX8) and repression of oncogenic drivers like *RUNX2* [118]. Epigenetically, TRβ interacts with chromatin remodeling complexes and coregulators (e.g., BRD4, LSD1, HDAC1/2), suggesting that combining TRβ agonists with epigenetic inhibitors may synergistically reverse dedifferentiation and restore tumor suppressive programs [119]. In vivo, GC-1 monotherapy suppressed ATC xenograft growth as effectively as sorafenib, with combination therapies further reducing proliferation markers like Ki-67 [116]. These findings underscore TRβ agonism as a promising strategy to reverse therapeutic resistance and redifferentiate aggressive thyroid cancers(Figure 2).

## 3. Clinical Correlation Between Thyroid Function Status and Thyroid Cancer

### 3.1. Graves’ Disease

Graves’ disease is an autoimmune disorder that occurs more frequently in populations with sufficient iodine intake. In such regions, it accounts for approximately 70–80% of all hyperthyroidism cases [120]. The disease is primarily driven by thyroid-stimulating hormone receptor antibodies (TSAb), which activate TSHR, leading to uncontrolled thyroid hormone production. Common clinical manifestations include hyperthyroidism and diffuse goiter, with orbital tissue swelling (orbitopathy) occurring in about 25% of patients [121].

#### 3.1.1. Graves and Nodules

A retrospective analysis of 557 patients undergoing surgery for Graves’ disease between 1991 and 1997 showed that 25.1% had preoperative nodules, and 15.0% were diagnosed with thyroid cancer [122]. Similarly, a study by Nazli Gülsoy Kırnap et al., involving 182 patients, reported thyroid nodules in 38% of individuals with Graves’ disease [123]. By comparison, the prevalence of palpable thyroid nodules in iodine-sufficient regions is approximately 5% in women and 1% in men [124], indicating a markedly higher frequency of nodularity in Graves’ patients. A meta-analysis by Papanastasiou et al., of seven studies including 2582 patients, found that the prevalence of TC was 22.2% in patients with nodules versus 5.1% in those without, yielding an odds ratio (OR) of 5.3 (95% CI: 2.4–11.6). This indicates a 5.3-fold increased risk of thyroid cancer in patients with thyroid nodules [125].

#### 3.1.2. Graves and Cancer

Compared to autonomous functioning thyroid nodules (AFTNs), thyroid cancers associated with Graves’ disease demonstrate greater aggressiveness, often presenting with multifocality, local invasion, metastasis, and higher recurrence rates [126]. The molecular basis for this aggressive phenotype involves the constitutive activation of TSHR by TSAbs, which occurs despite low circulating TSH levels in these patients [127]. This persistent TSHR signaling drives aberrant thyroid cell proliferation and impaired differentiation [126,128]. Furthermore, TSAbs have been shown to directly promote the growth and metastatic potential of thyroid cancer cells [126,129,130].

### 3.2. Potential Risks of Hypothyroidism

Hashimoto’s thyroiditis (HT), also referred to as chronic lymphocytic thyroiditis, is an autoimmune thyroid disorder [131] characterized primarily by painless goiter and potential progression to hypothyroidism [132]. Its pathogenesis involves a breakdown of immune tolerance, mediated by mechanisms such as CD8^+^ T cell cytotoxicity, interferon-gamma (IFN-γ) signaling, and anti-thyroid antibodies [133]. A large retrospective study of 64,628 patients who underwent thyroid surgery revealed a significantly higher rate of thyroid cancer—predominantly PTC—in those with histological confirmation of HT compared to non-HT patients (42.6% vs. 28.0%). Even after adjusting for age and sex, HT remained an independent risk factor for thyroid cancer (OR = 1.6) [134].

Congenital hypothyroidism (CH) results from developmental defects of the thyroid gland or severe iodine deficiency during gestation, leading to inadequate synthesis and secretion of T3/T4 and potentially causing intellectual disability in affected infants [135]. CH may confer a slightly elevated risk of DTC [130]. Proposed molecular mechanisms include impaired protein folding [136] and persistently elevated TSH levels, which may act synergistically with BRAF mutations, H_2_O_2_ accumulation, and DNA damage to promote carcinogenesis [137].

Subclinical hypothyroidism (SCH) represents an early phase of thyroid dysfunction, defined by elevated serum thyroid-stimulating hormone (TSH) levels while free T3 and T4 concentrations remain within normal reference intervals. A retrospective cohort study by Li et al., involving 13,717 patients who underwent surgery for papillary thyroid carcinoma between 2008 and 2017, demonstrated that SCH was an independent risk factor for extrathyroidal extension (OR = 1.168, 95% CI: 1.028–1.327, *p* = 0.017) [138].

#### Elevated TSH Promotes Cancer Development

In a meta-analysis of 28 studies encompassing 42,032 individuals and 5786 thyroid cancer cases, Donald S. A. McLeod et al. reported that serum TSH levels above 2.5 mIU/L were associated with a 1.8-fold increased risk of thyroid cancer compared to levels below 1.0 mIU/L (OR = 1.8, 95% CI: 1.3–2.5) [139].In an experimental model, Ohshima et al. induced hypothyroidism in rats using propylthiouracil (PTU) and administered N-2-hydroxypropylnitrosamine (DHPN) as a carcinogen. Within six months, the rats developed diffuse thyroid hyperplasia accompanied by markedly elevated serum TSH. By 52 weeks, the incidence of thyroid tumors was significantly higher in the DHPN + PTU group (92%) than in the DHPN-only group (32%), demonstrating that PTU potentiates DHPN-induced thyroid carcinogenesis via TSH-mediated mechanisms [140].

Under high TSH stimulation, NADPH oxidase is activated, resulting in a fourfold increase in reactive oxygen species (ROS) within thyroid cells. This oxidative stress causes DNA damage, cumulative gene mutations, and elevates H-Ras mutation rates by 5-fold [140]. Additionally, TSH activates the PI3K/Akt pathway, promoting phosphorylation of p70S6K, which enhances protein synthesis and suppresses the expression of the pro-apoptotic protein Bad [141]. TSH also binds to the TSHR, activating Gsα and increasing adenylate cyclase activity, which stimulates the cAMP/PKA pathway. This cascade upregulates Cyclin D1 transcription and protein expression, thereby driving cell proliferation [140].

## 4. TSH Suppression Therapy

TSH and TSHR, a member of the G protein-coupled receptor (GPCR) superfamily, play a central role in regulating thyroid hormone synthesis and secretion. TSH signaling through the cAMP/PKA pathway mediates critical processes such as iodine uptake, thyroperoxidase (TPO) activity regulation, and hydrogen peroxide production. Under physiological conditions, the thyroid gland secretes approximately 80–100 μg of T4 and 3–6 μg of T3 daily, with T3 constituting the biologically active form, accounting for 80–85% of circulating thyroid hormone activity. These hormones regulate essential bodily functions, including metabolic rate and thermogenesis [142].

TSH suppression therapy is a standard adjuvant treatment for patients with DTC following thyroidectomy. The core mechanism involves oral administration of levothyroxine to raise serum thyroid hormone levels, which suppresses pituitary-derived TSH secretion via negative feedback inhibition. By maintaining TSH at target concentrations, this approach mitigates the proliferative stimulus on any residual thyroid cancer cells, thereby reducing the risk of disease recurrence [143].

### 4.1. Risk-Stratified Targets and Clinical Outcomes

The 2015 American Thyroid Association (ATA) guidelines recommend a dynamic, risk-adapted approach to TSH suppression [124]. This stratification is crucial because the clinical benefit of TSH suppression is proportional to the initial risk of recurrence, while the risks of adverse effects remain significant across all groups.

High-risk patients: For those with persistent structural disease, maintaining TSH < 0.1 mU/L is associated with the greatest reduction in recurrence and disease-specific mortality. The benefit of aggressive suppression in this group generally outweighs the risks.

Intermediate-risk patients: For this group, a TSH target of 0.1–0.5 mU/L is recommended. Studies suggest that this moderate suppression provides a significant reduction in recurrence risk without incurring the full spectrum of severe side effects associated with deeper suppression.

Low-risk patients: For patients who have undergone lobectomy or have an excellent response to initial therapy, maintaining TSH within the physiologic range (0.5–2.0 mU/L) is sufficient. 

### 4.2. Risk of Overtreatment

Excessive suppression may lead to adverse effects, including impaired cardiovascular function [144], reduced bone mineral density and symptoms of subclinical hyperthyroidism [145,146].

Thyroid hormone increases the sensitivity of the beta-adrenergic receptor complex and directly alters the expression and activity of cardiac ion channels. This combination lowers the threshold for atrial fibrillation, and meta-analyses show a significant increase in risk (RR = 1.52) [147]. In postmenopausal women, who lack the protective effect of estrogen, TSH suppression leads to a significant reduction in lumbar spine BMD (WMD −0.03 g/cm^2^) [148]. The rate of loss is progressive, with the time for 10% of patients to develop osteoporosis shortening from 85 months with mild osteopenia to just 15 months with severe osteopenia [149]. Furthermore, excessive TSH suppression frequently results in exogenous subclinical thyrotoxicosis, characterized by undetectable serum TSH concentrations with normal T3/T4 concentrations [150].

### 4.3. Therapeutic Limitations

No data indicates that low serum thyrotropin levels in patients with low-risk tumors following thyroidectomy confer clinical benefit [151]. The management of TSH suppression therapy currently relies on broad risk categories. There is a significant lack of precise biomarkers to predict an individual patient’s response to suppression or their susceptibility to its long-term adverse effects, leading to considerable interpatient variability in outcomes and quality of life.

## 5. Conclusions and Future Perspective

This review systematically examined the complex and dualistic roles of thyroid hormones and TSH in the pathogenesis, progression, and treatment of thyroid cancer. We highlighted the critical involvement of both genomic (TR-mediated) and non-genomic (integrin αvβ3-mediated) signaling pathways, underscoring how dysregulation at the hormonal, receptor, or effector level can drive oncogenic processes.

Despite the clarity afforded by this molecular model, its clinical translation faces considerable challenges. A major limitation lies in the absence of standardized assays for key biomarkers—such as TRβ mutations and integrin αvβ3 expression—in routine diagnostics. Furthermore, the application of cornerstone adjuvant therapies like TSH suppression remains imprecise. Although current risk-stratified approaches represent an advancement in this field, they still fail to adequately capture interpatient heterogeneity due to the lack of reliable biomarkers to predict individual therapeutic response and susceptibility to adverse effects, thus limiting optimization of the risk–benefit profile.

Future efforts should focus on leveraging these mechanistic insights to develop targeted therapies and precision biomarker integration, so that we can exploit the dual nature of thyroid hormone signaling.

## Figures and Tables

**Figure 1 biomedicines-13-02552-f001:**
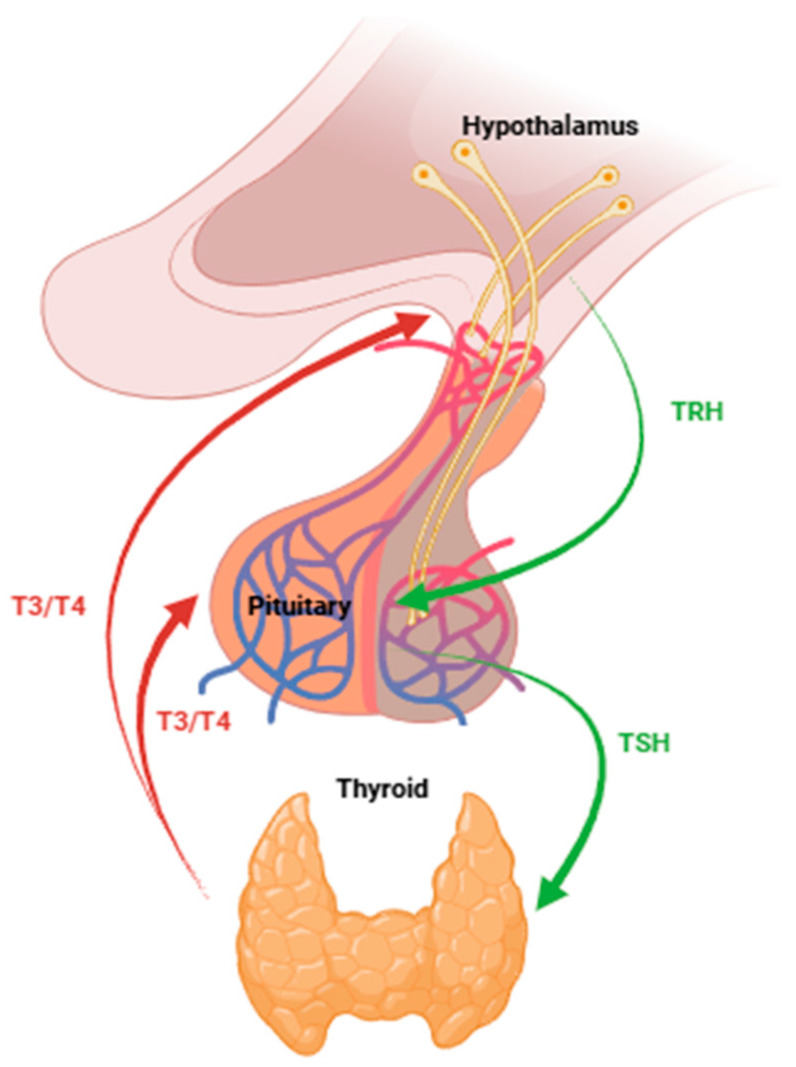
Regulation of the thyroid hormone axis. The hypothalamus releases TRH, which stimulates thyrotrope cells in the pituitary gland to synthesize and release TSH. TSH then acts on the thyroid gland to promote the production and secretion of thyroid hormones T3 and T4. T3 and T4 exert negative feedback inhibition on both the pituitary and the hypothalamus to suppress the release of TSH and TRH, respectively, completing the feedback loop and maintaining hormonal homeostasis. Green arrows represent stimulatory pathways; red arrows represent inhibitory feedback.

**Figure 2 biomedicines-13-02552-f002:**
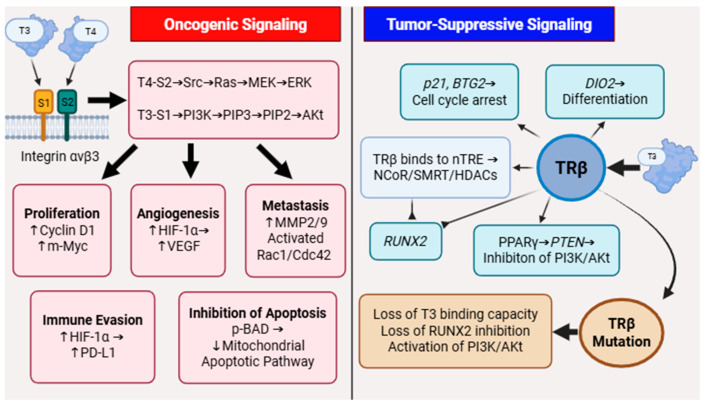
The Dual Role of T3/T4 in Oncogenic and Tumor-Suppressive Pathways. Oncogenic Signaling: Initiated by inputs such as integrin αvβ3, activation of the Src/Ras/MEK/ERK and PI3K/Akt pathways promotes tumorigenesis by driving proliferation (via Cyclin D1, c-Myc), angiogenesis (via HIF-1α/VEGF), metastasis (via MMP2/9, Rac1/Cdc42), immune evasion (via HIF-1α/PD-L1), and inhibition of apoptosis (via p-BAD). Tumor-Suppressive Signaling: TRβ exerts its protective effects by inducing cell cycle arrest genes (p21, BTG2), promoting differentiation (via DIO2), recruiting transcriptional corepressor complexes (NCoR/SMRT/HDACs), and inhibiting key oncogenic drivers like RUNX2 and the PI3K/Akt pathway (via PPARγ/PTEN). The loss of TRβ function, through mutation or loss of T3 binding, disrupts this balance, leading to unleashed oncogenic signaling and disease progression.

**Table 1 biomedicines-13-02552-t001:** Selected genes relevant to thyroid cancer whose expression is subject to T3-αvβ3 pathway.

Gene	Effect	References
Promoting proliferation genes		
*CCND1*	↑	[50]
Apotosis		
*Bcl-2*	↑	[51]
*XIAP*	↑	[52]
Angiogenesis genes		
*VEGF*	↑	[53]
*bFGF*	↑	[54]
*PDGF*	↑	[55]
*HIF-1α*	↑	[56]
*NOS2*	↑	[57]
Tumor metastasis genes		
*SNAL1*	↑	[58]
*MMPs*	↑	[59]

“↑” indicates a promoting effect. bFGF, basic fibroblast growth factor; Bcl-2, B-cell lymphoma 2; CCND1, cyclin D1; HIF-1α, hypoxia-inducible factor 1-alpha; MMPs, matrix metalloproteinases; NOS2, nitric oxide synthase 2; PDGF, platelet-derived growth factor; SNAI1, snail family transcriptional repressor 1; VEGF, vascular endothelial growth factor; XIAP, X-linked inhibitor of apoptosis.

## Data Availability

Not applicable.

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
