# Peer review of "Molecular Mechanisms of Thyroid Hormone Signaling in Thyroid Cancer: Oncogenesis, Progression, and Therapeutic Implications"

_biomedicines, 2025, doi:10.3390/biomedicines13102552_

Round 1
Reviewer 1 Report
Comments and Suggestions for Authors
Thank you for the opportunity to review the manuscript “Molecular Mechanisms of Thyroid Hormone Signaling in Thyroid Cancer: Oncogenesis, Progression, and Therapeutic Implications” by Changhao Zhou from China. It discusses molecular mechanisms in thyroid cancer. However, in my opinion there is lack of a structured literature review. Moreover some paragraphs and statements are not clear. I would recommend to reconsider the aim, methodology and presentation of the article and resubmit the manuscript.
There are four types of thyroid cancer. Each of these cancers has different mechanisms and etiopathogenesis. For example, medullary thyroid cancer has a familial inheritance, and the RET proto-oncogene plays importantant role in its etiopathogenesis. Authors should detail which cancers these mechanisms and TSH stimulation play a role in their etiopathogenesis.
The some genetic mechanisms causing thyroid cancer should be mentioned in the discussion section. For example, many factors such as radiation, diet, stress…. may play a role in the etiology of thyroid cancers through epigenetic factors such as methylation, histone modification, and mRNA.
Lack of separate paragraphs regarding papillary, follicular, medullary and anaplastic thyroid cancers
Some diagnosting limitations to this study should be emphasized.
English revision and editing by a native speaker is recommended.
English revision and editing by a native speaker is recommended.
Author Response
Dear Editors and Reviewers,
We appreciate opportunity to revise our manuscript titled “Molecular Mechanisms of Thyroid Hormone Signaling in Thyroid Cancer: Oncogenesis, Progression, and Therapeutic Implications” and are thankful for the insightful comments provided. Those comments are really useful for revising my paper. We provided detailed responses to each of the reviewers’ comments. Revisions are marked using the “Track Changes” function. The corrections in the paper and the response are as following:
Comments 1. However, in my opinion there is lack of a structured literature review. Moreover some paragraphs and statements are not clear. I would recommend to reconsider the aim, methodology and presentation of the article and resubmit the manuscript.
Response: We are profoundly grateful for your time and for providing such critical and constructive feedback on our manuscript. We take your comments regarding the lack of a structured literature review, unclear paragraphs, and the need to reconsider the article's aim and methodology with the utmost seriousness.
In response, we have not merely revised but have fundamentally restructured and rewritten the manuscript to address these core concerns. We now explicitly define the specific research questions the article seeks to answer, setting clear boundaries for the literature covered and stating the unique perspective or synthesis this review provides to the field.
Comments 2. There are four types of thyroid cancer. Each of these cancers has different mechanisms and etiopathogenesis. For example, medullary thyroid cancer has a familial inheritance, and the RET proto-oncogene plays importantant role in its etiopathogenesis. Authors should detail which cancers these mechanisms and TSH stimulation play a role in their etiopathogenesis.
Response: We thank you for this critical insight. We agree that a more precise delineation of the mechanisms according to thyroid cancer subtypes is essential. We have thoroughly revised the manuscript to specify the cancer type (line 76-88).
The revised content is as follows:
DTCs are frequently driven by mutations in the MAPK pathway, typically at BRAF or RAS oncoproteins[16]. MTC originated from parafollicular C-cells and is strongly related to germline or somatic mutations in the RET proto-oncogene and lower-frequently associated with KRAS and BRAF . And its biological behavior, genetic basis, and lack of responsiveness to TSH are fundamentally distinct.[17]. ATC is derived from follicular thyroid cells and carries the highest risk of death, yet accounts for only a small fraction of all TC cases[18]. This review will focus primarily on DTC, given its high prevalence and the well-established role of thyroid hormone signaling in its pathogenesis, though relevant contrasts with MTC and ATC will be noted where appropriate.
References:
- Leandro-García LJ, Landa I. Mechanistic Insights of Thyroid Cancer Progression. ENDOCRINOLOGY 2023; 164(9).
- Prabhash K, Saldanha E, Patil V, et al. RET Alterations Differentiate Molecular Profile of Medullary Thyroid Cancer. JCO PRECIS ONCOL 2024; 8: e2300622.
- Bible KC, Kebebew E, Brierley J, et al. 2021 American Thyroid Association Guidelines for Management of Patients with Anaplastic Thyroid Cancer. THYROID 2021; 31(3): 337-86
Comments 3. The some genetic mechanisms causing thyroid cancer should be mentioned in the discussion section. For example, many factors such as radiation, diet, stress…. may play a role in the etiology of thyroid cancers through epigenetic factors such as methylation, histone modification, and mRNA.
Response: We thank the reviewer for this excellent suggestion to enhance the discussion section. We have added epigenetic modifications in thyroid tumorigenesis(line 88-99).
The revised content is as follows:
Beyond genetic alterations, emerging evidence underscores the importance of epigenetic modifications in thyroid tumorigenesis. These mechanisms primarily involve hypermethylation of DNA in promoter CpG islands and post-translational modifications of histones (such as deacetylation and methylation), leading to transcriptional silencing of key genes [19-21]. In thyroid cancer, this results in loss of thyroid-specific function, such as impaired radioactive iodine uptake due to silencing of the NIS and TSHR genes[22]. These mechanisms can promote uncontrolled cell proliferation through inactivation of tumor suppressor genes like RASSF1A, PTEN, and RAP1GAP[23-25]. The reversibility of these epigenetic marks offers promising therapeutic avenues. Strategies aim to reverse gene silencing through demethylating agents and histone deacetylase inhibitors (HDACi) to restore differentiation and suppress tumor growth, particularly in advanced, refractory cases[26].
References:
- Kondo T, Asa SL, Ezzat S. Epigenetic dysregulation in thyroid neoplasia. ENDOCRIN METAB CLIN 2008; 37(2): 389-400.
- Poke FS, Qadi A, Holloway AF. Reversing aberrant methylation patterns in cancer. CURR MED CHEM 2010; 17(13): 1246-54.
- Chi P, Allis CD, Wang GG. Covalent histone modifications--miswritten, misinterpreted and mis-erased in human cancers. NAT REV CANCER 2010; 10(7): 457-69.
- Gérard A, Daumerie C, Mestdagh C, et al. Correlation between the loss of thyroglobulin iodination and the expression of thy-roid-specific proteins involved in iodine metabolism in thyroid carcinomas. J CLIN ENDOCR METAB 2003; 88(10): 4977-83.
- Alvarez-Nuñez F, Bussaglia E, Mauricio D, et al. PTEN promoter methylation in sporadic thyroid carcinomas. THYROID 2006; 16(1): 17-23.
- Faam B, Ghaffari MA, Khorsandi L, et al. RAP1GAP Functions as a Tumor Suppressor Gene and Is Regulated by DNA Methylation in Differentiated Thyroid Cancer. CYTOGENET GENOME RES 2021; 161(5): 227-35.
- Kunstman JW, Korah R, Healy JM, Prasad M, Carling T. Quantitative assessment of RASSF1A methylation as a putative molecular marker in papillary thyroid carcinoma. SURGERY 2013; 154(6): 1255-61, 1261-2.
- Catalano MG, Poli R, Pugliese M, Fortunati N, Boccuzzi G. Emerging molecular therapies of advanced thyroid cancer. MOL ASPECTS MED 2010; 31(2): 215-26.
Comments 4. Some diagnosting limitations to this study should be emphasized.
Response: Thanks for your valuable comment. We have now incorporated a detailed discussion on therapeutic limitations (line 562-568).
The revised content is as follows:
Though, no data is indicating that low serum thyrotropin levels in patients with low-risk tumors following thyroidectiomy confer clinical benefit[159]. The management of TSH suppression therapy currently relies on broad risk categories. There is a significant lack of precise biomarkers to predict an individual patient's response to suppression or their susceptibility to its long-term adverse effects, leading to considerable interpatient variability in outcomes and quality of life.
References:
- Haugen BR. 2015 American Thyroid Association Management Guidelines for Adult Patients with Thyroid Nodules and Differentiated Thyroid Cancer: What is new and what has changed? CANCER-AM CANCER SOC 2017; 123(3): 372-81.
Comments 5. English revision and editing by a native speaker is recommended.
Response: Thanks for your suggestion. The manuscript has undergone thorough English language editing by a native speaker to correct grammatical errors, improve sentence structure, and enhance overall readability.

Reviewer 2 Report
Comments and Suggestions for Authors
The manuscript titled, Molecular Mechanisms of Thyroid Hormone Signaling in Thyroid Cancer: Oncogenesis, Progression, and Therapeutic Implications is well-structured and scientifically rigorous, providing a thorough molecular overview of thyroid hormone signaling in thyroid cancer. However, improving the sections on clinical controversies, therapeutic innovations, and future directions will significantly boost its relevance and attract more readers.
Major Concerns
- The review is comprehensive, but much of the material summarizes already well-established mechanisms (integrin αvβ3, TRβ mutations, PI3K/Akt, MAPK/ERK). The paper would benefit from highlighting what new insights this review provides compared to existing reviews.
- Limited critical evaluation of controversies in the field. For example, whether elevated TSH directly promotes tumor progression versus being a surrogate marker is still debated.
- The manuscript is heavily weighted toward molecular signaling, with detailed descriptions of pathways, but the clinical implications (e.g., TSH suppression therapy, prognostic relevance of TRβ mutations) are discussed only briefly. A more balanced discussion would strengthen translational relevance.
- No graphical summary highlighting the “dual role” of thyroid hormones (pro-tumorigenic vs. tumor-suppressive). This would aid clarity.
- The discussion of TSH suppression therapy primarily references ATA guidelines, but it does not sufficiently cover ongoing debates about the risks of overtreatment—such as atrial fibrillation and bone loss compared to its limited benefit in low-risk differentiated thyroid cancer.
- A systematic comparison of high-, intermediate, and low-risk patient outcomes under suppression therapy is lacking.
- Although the title promises “therapeutic implications,” the manuscript devotes little space to novel therapies. For example, tetrac/NDAT (thyrointegrin αvβ3 antagonists) and TRβ agonists are briefly mentioned but not critically appraised. No mention of ongoing clinical trials that target thyroid hormone pathways, which would make the paper more future-looking.
Minor Concerns
- Some sentences are overly long and dense (e.g., sections on integrin αvβ3 signaling). These should be broken down for readability.
- Occasional typographical inconsistencies (e.g., spacing before citations).
- Figure legends are missing the full form of all the abbreviations used in the figures.
- Some pathways (MAPK/ERK, PI3K/Akt) are described multiple times in different sections (integrin αvβ3, TRβ mutations, TSH signaling), leading to repetition. These could be consolidated.
Author Response
Dear reviewer,
We appreciate the constructive feedback provided by the editor and reviewers on our manuscript. Your comments are crucial for us to improve the paper. We have carefully considered all comments and made the necessary revisions on the manuscript. The following provides detailed responses to each of the comments. Revisions are marked using the “Track Changes” function. The corrections in the paper and the response are as following:
- Major Concerns:
Comments 1. The review is comprehensive, but much of the material summarizes already well-established mechanisms (integrin αvβ3, TRβ mutations, PI3K/Akt, MAPK/ERK). The paper would benefit from highlighting what new insights this review provides compared to existing reviews.
Response: Thank you for this insightful comment. We have now revised the conclusion sections to more explicitly state this integrative and network-based perspective as our review's unique contribution.
Comments 2. Limited critical evaluation of controversies in the field. For example, whether elevated TSH directly promotes tumor progression versus being a surrogate marker is still debated.
Response: Thank you for raising this critical point. We have added a deeper discussion of on going controversies in the TSH suppression therapy section.
Though, no data is indicating that low serum thyrotropin levels in patients with low-risk tumors following thyroidectiomy confer clinical benefit[159].
- Haugen BR. 2015 American Thyroid Association Management Guidelines for Adult Patients with Thyroid Nodules and Differentiated Thyroid Cancer: What is new and what has changed? CANCER-AM CANCER SOC 2017; 123(3): 372-81.
Comments 3. The manuscript is heavily weighted toward molecular signaling, with detailed descriptions of pathways, but the clinical implications (e.g., TSH suppression therapy, prognostic relevance of TRβ mutations) are discussed only briefly. A more balanced discussion would strengthen translational relevance.
Response: Thanks for suggestion. We have balanced our discussion, with more detailed discussion about TSH suppression therapy and prognostic significance of TRβ mutations (1.2.6).
The revised content is as follows:
1.2.6 Prognostic Significance of TRβ Mutations
The presence of TRβ mutations or loss of expression is a significant marker of poor prognosis in thyroid cancer. Functionally impaired TRβ mutants, such as the dominant-negative TRβPV, not only lose transcriptional activity but also drive tumor progression through extranuclear signaling mechanisms[115]. For instance, TRβPV physically interacts with the p85α regulatory subunit of PI3K, leading to constitutive activation of the PI3K/Akt pathway, which enhances cell proliferation and suppresses apoptosis[116-117].
Moreover, TRβPV disrupts proteasomal degradation pathways, resulting in aberrant accumulation of oncoproteins such as PTTG and β-catenin[118]. This leads to mitotic abnormalities, chromosomal instability, and epithelial–mesenchymal transition (EMT)-like phenotypes, further driving dedifferentiation and metastatic behavior[119].
In summary, TRβ mutations enhance tumor invasiveness, including extraluminal invasion, vascular invasion, and distant metastasis[120]. Furthermore, these mutations may confer resistance to conventional therapies such as TSH suppression and radioactive iodine treatment, highlighting their potential as prognostic biomarkers and therapeutic targets in advanced thyroid cancer[121].
References:
- Suzuki H, Willingham MC, Cheng S. Mice with a mutation in the thyroid hormone receptor beta gene spontaneously develop thyroid carcinoma: a mouse model of thyroid carcinogenesis. THYROID 2002; 12(11): 963-9.
- Furuya F, Hanover JA, Cheng S. Activation of phosphatidylinositol 3-kinase signaling by a mutant thyroid hormone beta receptor. P NATL ACAD SCI USA 2006; 103(6): 1780-5.
- Furuya F, Guigon CJ, Zhao L, Lu C, Hanover JA, Cheng S. Nuclear receptor corepressor is a novel regulator of phosphatidylinositol 3-kinase signaling. MOL CELL BIOL 2007; 27(17): 6116-26.
- Ying H, Furuya F, Zhao L, et al. Aberrant accumulation of PTTG1 induced by a mutated thyroid hormone beta receptor inhibits mitotic progression. J CLIN INVEST 2006; 116(11): 2972-84.
- Guigon CJ, Zhao L, Lu C, Willingham MC, Cheng S. Regulation of beta-catenin by a novel nongenomic action of thyroid hormone beta receptor. MOL CELL BIOL 2008; 28(14): 4598-608.
- Guigon CJ, Cheng S. Novel non-genomic signaling of thyroid hormone receptors in thyroid carcinogenesis. MOL CELL ENDOCRINOL 2009; 308(1-2): 63-9.
- Aoyama M, Yamasaki S, Tsuyuguchi M. A case of resistance to thyroid hormone diagnosed after total thyroidectomy for thyroid cancer. J MED INVESTIG 2015; 62(3-4): 268-71.
Comments 4. No graphical summary highlighting the “dual role” of thyroid hormones (pro-tumorigenic vs. tumor-suppressive). This would aid clarity.
Response: We sincerely appreciate your valuable comment. We have modified the original Figure 2 to emphasize the dual role of thyroid hormones. This figure is divided into two sections: the left side shows tumor-promoting signals, while the right side depicts tumor-suppressing signals.
Figure 2. The Dual Role of T3/T4 in Oncogenic and Tumor-Suppressive Pathways.
Comments 5. The discussion of TSH suppression therapy primarily references ATA guidelines, but it does not sufficiently cover ongoing debates about the risks of overtreatment—such as atrial fibrillation and bone loss compared to its limited benefit in low-risk differentiated thyroid cancer.
Response: Thanks for the valuable feedback. We have added the risks of overtreatment of TSH suppression therapy.
The revised content is as follows:
Excessive suppression may lead to adverse effects including impaired cardiovascular function[153], reduced bone mineral density[154] and symptoms of sub-clinical hyperthyroidism[155].
Thyroid hormone increases the sensitivity of the beta-adrenergic receptor complex and directly alters the expression and activity of cardiac ion channels. This combination lowers the threshold for atrial fibrillation, and meta-analyses show a significant increase in risk (RR = 1.52)[156]. In postmenopausal women, who lack the protective effect of estrogen, TSH suppression leads to a significant reduction in lumbar spine BMD (WMD -0.03 g/cm²) [157]. The rate of loss is progressive, with the time for 10% of patients to develop osteoporosis shortening from 85 months with mild osteopenia to just 15 months with severe osteopenia[158]. Futhermore, excessive TSH suppression frequently results in exogenous subclinical thyrotoxicosis, characterized by undetectable serum TSH concentrations with normal T3/T4 concentrations[159].
References:
- Biondi B, Fazio S, Cuocolo A, et al. Impaired cardiac reserve and exercise capacity in patients receiving long-term thyrotropin suppressive therapy with levothyroxine. J CLIN ENDOCR METAB 1996; 81(12): 4224-8.
- Lamartina L, Durante C, Lucisano G, et al. Are Evidence-Based Guidelines Reflected in Clinical Practice? An Analysis of Prospectively Collected Data of the Italian Thyroid Cancer Observatory. THYROID 2017; 27(12): 1490-7.
- Do Cao C, Wémeau JL. Risk-benefit ratio for TSH- suppressive Levothyroxine therapy in differentiated thyroid cancer. ANN ENDO-CRINOL-PARIS 2015; 76(1, Supplement 1): 1S-47S.
- Yang X, Guo N, Gao X, Liang J, Fan X, Zhao Y. Meta-analysis of TSH suppression therapy and the risk of cardiovascular events after thyroid cancer surgery. FRONT ENDOCRINOL 2022; 13: 991876.
- Ku EJ, Yoo WS, Lee EK, et al. Effect of TSH Suppression Therapy on Bone Mineral Density in Differentiated Thyroid Cancer: A Systematic Review and Meta-analysis. J CLIN ENDOCR METAB 2021; 106(12): 3655-67.
- Park H, Park J, Yoo H, et al. Bone-density testing interval and transition to osteoporosis in differentiated thyroid carcinoma patients on TSH suppression therapy. CLIN ENDOCRINOL 2022; 97(1): 130-6.
Comments 6. A systematic comparison of high, intermediate, and low-risk patient outcomes under suppression therapy is lacking.
Response: Thank you for the suggesstion. We added risk stratified targets and clinical outcomes (line 531-546).
The revised content is as follows:
3.1 Risk-Stratified Targets and Clinical Outcomes
The 2015 American Thyroid Association (ATA) guidelines recommend a dynamic, risk-adapted approach to TSH suppression[151]. This stratification is crucial because the clinical benefit of TSH suppression is proportional to the initial risk of recurrence, while the risks of adverse effects remain significant across all groups.
High-Risk Patients: For those with persistent structural disease, maintaining TSH <0.1 mU/L is associated with the greatest reduction in recurrence and disease-specific mortality. The benefit of aggressive suppression in this group generally outweighs the risks.
Intermediate-Risk Patients: For this group, a TSH target of 0.1-0.5 mU/L is recommended. Studies suggest that this moderate suppression provides a significant reduction in recurrence risk without incurring the full spectrum of severe side effects associated with deeper suppression.
Low-Risk Patients: For patients who have undergone lobectomy or have an excellent response to initial therapy, maintaining TSH within the physiologic range (0.5-2.0 mU/L) is sufficient.
References:
- Haugen BR, Alexander EK, Bible KC, et al. 2015 American Thyroid Association Management Guidelines for Adult Patients with Thyroid Nodules and Differentiated Thyroid Cancer: The American Thyroid Association Guidelines Task Force on Thyroid Nodules and Differentiated Thyroid Cancer. THYROID 2016; 26(1): 1-133.
Comments 7. Although the title promises “therapeutic implications,” the manuscript devotes little space to novel therapies. For example, tetrac/NDAT (thyrointegrin αvβ3 antagonists) and TRβ agonists are briefly mentioned but not critically appraised. No mention of ongoing clinical trials that target thyroid hormone pathways, which would make the paper more future-looking.
Response: We appreciate your comment. We have expanded the discussion on novel targeted therapeutic strategies against MAPK/ERK and PI3K/Akt pathways (line 285-303) and preclinical machanism studies of TRβ in antitumor therapy (line 412-424).
The revised contents are as follows:
1.1.8 Novel targeted therapeutic strategies against MAPK/ERK and PI3K/Akt Pathways
For radioactive iodine-refractory and advanced thyroid cancer, targeted therapies against the MAPK and PI3K/Akt pathways are a treatment cornerstone[79]. Clinical evidence shows that inhibiting key nodes in these pathways effectively suppresses tumor progression and improves outcomes.
Within the MAPK pathway, the BRAF V6000E mutation is a pivotal therapeutic target. Selective BRAF V6000E inhibitors vemurafenib and dabrafenib have demonstrated significant antitumor activity in clinical trials[80]. For ATC, the combination of dabrafenib and the MEK inhibitor trametinib has shown breakthrough efficacy, achieving a 69% ORR in a phase II study and subsequently gaining FDA approval as the first effective targeted regimen for this disease[81].
Emerging research frontiers are increasingly centered on exploring synergistic combinations of targeted therapy and immunotherapy. Preclinical studies demonstrate that co-administration of BRAF V600E inhibitors with anti-PD-1/PD-L1 antibodies significantly suppresses ATC progression and remodels the tumor immune microenvironment toward an immunologically active state[82-83]. Although clinical validation is still evolving, this combined approach represents a pivotal strategy for overcoming therapeutic resistance and improving outcomes in refractory thyroid cancer.
References:
- Laha D, Nilubol N, Boufraqech M. New Therapies for Advanced Thyroid Cancer. FRONT ENDOCRINOL 2020; 11: 82.
- Brose MS, Cabanillas ME, Cohen EEW, et al. Vemurafenib in patients with BRAF(V600E)-positive metastatic or unresectable papillary thyroid cancer refractory to radioactive iodine: a non-randomised, multicentre, open-label, phase 2 trial. LANCET ONCOL 2016; 17(9): 1272-82.
- Subbiah V, Kreitman RJ, Wainberg ZA, et al. Dabrafenib and Trametinib Treatment in Patients With Locally Advanced or Metastatic BRAF V600-Mutant Anaplastic Thyroid Cancer. J CLIN ONCOL 2018; 36(1): 7-13.
- Brauner E, Gunda V, Vanden Borre P, et al. Combining BRAF inhibitor and anti PD-L1 antibody dramatically improves tumor re-gression and anti tumor immunity in an immunocompetent murine model of anaplastic thyroid cancer. Oncotarget 2016; 7(13): 17194-211.
- Gunda V, Gigliotti B, Ashry T, et al. Anti-PD-1/PD-L1 therapy augments lenvatinib's efficacy by favorably altering the immune microenvironment of murine anaplastic thyroid cancer. INT J CANCER 2019; 144(9): 2266-78.
1.2.7 Preclinical machanism studies of TRβ in antitumor therapy
Preclinical studies have established the potent tumor-suppressive mechanisms of TRβ agonists in ATC[122]. TRβ agonists GC-1 reprograms the transcriptome and epigenome, leading to reduced cell proliferation, metastasis, while promoting redifferentiation and apoptosis[123]. Key mechanisms include the induction of a tumor-suppressive gene network involving thyroid differentiation markers (e.g., NIS/SLC5A5, TG, PAX8) and repression of oncogenic drivers like RUNX2[124]. Epigenetically, TRβ interacts with chromatin remodeling complexes and coregulators (e.g., BRD4, LSD1, HDAC1/2), suggesting that combining TRβ agonists with epigenetic inhibitors may synergistically reverse dedifferentiation and restore tumor suppressive programs[125]. In vivo, GC-1 monotherapy suppressed ATC xenograft growth as effectively as sorafenib, with combination therapies further reducing proliferation markers like Ki-67[122]. These findings underscore TRβ agonism as a promising strategy to reverse therapeutic resistance and redifferentiate aggressive thyroid cancers.
References:
- Gillis NE, Cozzens LM, Wilson ER, et al. TRβ Agonism Induces Tumor Suppression and Enhances Drug Efficacy in Anaplastic Thyroid Cancer in Female Mice. ENDOCRINOLOGY 2023; 164(10).
- Pourvali K, Shimi G, Ghorbani A, Shakery A, Shirazi FH, Zand H. Selective thyroid hormone receptor beta agonist, GC-1, is capable to reduce growth of colorectal tumor in syngeneic mouse models. J RECEPT SIG TRANSD 2022; 42(5): 495-502.
- Bolf EL, Gillis NE, Davidson CD, et al. Thyroid Hormone Receptor Beta Induces a Tumor-Suppressive Program in Anaplastic Thyroid Cancer. MOL CANCER RES 2020; 18(10): 1443-52.
- Rustad JL, Gillis NE, Lignos J, Bright KA, Frietze S, Carr FE. Epigenomic Modulators and Thyroid Hormone Receptor β Agonists: A New Paradigm for Tumor Suppression in Thyroid Cancer. ENDOCRINOLOGY 2025; 166(9).
- Minor Concerns:
Comments 1. Some sentences are overly long and dense (e.g., sections on integrin αvβ3 signaling). These should be broken down for readability.
Response: Thank for your feedback. We have thoroughly reviewed the manuscript, with particular attention to the section on integrin αvβ3 signaling and other dense passages. The problematic long sentences have been broken down into shorter, clearer sentences. We believe these revisions have significantly enhanced the overall clarity and flow of the text.
Comments 2. Occasional typographical inconsistencies (e.g., spacing before citations).
Response: We thank you for the valuable comment. These issues have been addressed accordingly.
Comments 3. Figure legends are missing the full form of all the abbreviations used in the figures.
Response: Thanks for the valuable suggestion. We have added the abbreviated terms’ interpretation in manuscript supplementary material as follows:
- Abbreviation terms
TH, thyroid hormone; TSH, thyroid-stimulating hormone; TRH, thyrotropin-releasing hormone; TSHR, thyroid-stimulating hormone receptor; HPT axis, hypothalamic--pituitary--thyroid axis; ASR, age-standardized incidence rate; MAPK, mitogen-activated protein kinase; BRAF, v-Raf murine sarcoma viral oncogene homolog B; RAS, rat sarcoma virus; RET, rearranged during Transfection; CpG, cytosine-phosphate-guanine; HDACi, histone deacetylase inhibitors; NIS, sodium iodide symporter; PTEN, phosphatase and tensin homolog; TRE, thyroid hormone response element; PI3K, phosphoinositide 3-kinase; Akt, protein kinase B; ERK, extracellular signal-regulated kinase; PIP2, phosphatidylinositol 4,5-bisphosphate; PIP3, phosphati-dylinositol (3,4,5)-trisphosphate; mTOR, mammalian target of rapamycin; FOXO, Forkhead box O; GSK-3β, glycogen synthase kinase-3 beta; CAM, chick chorioallantoic membrane; ROS, reactive oxygen species; HIF-1α, hypoxia-inducible factor 1-alpha; VEGF, vascular endothelial growth factor; FGF2, fibroblast growth factor 2; PDGF, platelet-derived growth factor; EGFR, epidermal growth factor receptor; MMP, matrix metalloproteinase; NOS2, nitric oxide synthase 2; TGF-β, transforming growth factor beta; Rho, Ras homolog; Rac1, Ras-related C3 botulinum toxin substrate 1; Cdc42, cell division cycle 42; GEFs, guanine nucleotide exchange factors; RV, resveratrol; BAD, BCL2-associated agonist of cell death; Bcl-2, B-cell lymphoma 2; PD-L1, programmed death-ligand 1; PD-1, programmed death-1; HREs, hypoxia-response elements; Tetrac, tetraiodothyroacetic acid; NDAT, nano-diaminotetrac; CCND1, Cyclin D1; XIAP, X-linked inhibitor of apoptosis; bFGF, basic fibroblast growth factor; SNAI1, snail family transcriptional repressor 1; LBD, ligand-binding domain; AEV, avian erythroblastosis virus; NCoR, nuclear receptor co-repressor; SMRT, silencing mediator for retinoid and thyroid hormone receptors; CBP, CREB-binding protein; DIO2, type 2 iodothyronine deiodinase; BTG2, B-cell translocation gene 2; ccRCC, clear cell renal cell carcinoma; HCC, hepatocellular carcinoma; PPARγ, peroxisome proliferator-Activated receptor gamma; PPRE, peroxisome proliferator response element; PP2A, protein phosphatase 2A; FAK, focal adhesion kinase; EMT, epithelial-mesenchymal transition; RUNX2, Runt-related transcription factor 2; nTREs, negative thyroid hormone response elements; PTTG, pituitary tumor-transforming gene; HT, Hashimoto's thyroiditis; CH, congenital hypothyroidism; SCH, subclinical hypothyroidism; PTU, propylthiouracil; DHPN, N-2-hydroxypropylnitrosamine; NADPH, nicotinamide adenine dinucleotide phosphate; Gsα, stimulatory G protein alpha subunit; BRD4, Bro-modomain-containing protein 4; LSD1, Lysine-specific demethylase 1; HDAC, histone deacetylase.
Comments 4. Some pathways (MAPK/ERK, PI3K/Akt) are described multiple times in different sections (integrin αvβ3, TRβ mutations, TSH signaling), leading to repetition. These could be consolidated.
Response: We would love to thank you for the insightful comments. We have corrected the errors in the revised manuscript (like line 372-373).

Round 2
Reviewer 1 Report
Comments and Suggestions for Authors
It is my pleasure to review the revision manuscript entitled “Molecular Mechanisms of Thyroid Hormone Signaling in Thyroid Cancer: Oncogenesis, Progression, and Therapeutic Implications” by Dr Chang Hao Zhou et al. The summary of the concerns are as follows:
- The objective of this study have been clearly explained.
- The study design is straightforward.
- English language fine. No issues detected.
- Overall, I would recommend publication of the manuscript because the amendments were made.
Reviewer 2 Report
Comments and Suggestions for Authors
I appreciate the authors for addressing all of my concerns! I strongly recommend publishing the current form of this manuscript.